# Long-Term Outcomes of Extra-Anatomic Femoro-Tibial Bypass Reconstructions in Chronic Limb-Threating Ischemia

**DOI:** 10.3390/jcm11051237

**Published:** 2022-02-24

**Authors:** Alexander Meyer, Evgenia Boxberger, Christian-Alexander Behrendt, Shatlyk Yagshyyev, Irina Welk, Werner Lang, Ulrich Rother

**Affiliations:** 1Department of Vascular Surgery, University Hospital Erlangen, Friedrich-Alexander University Erlangen-Nuremberg, Krankenhausstrasse 12, 91054 Erlangen, Germany; evgenia.boxberger@uk-erlangen.de (E.B.); shatlyk.yagshyyev@uk-erlangen.de (S.Y.); irina.welk@uk-erlangen.de (I.W.); werner.lang@uk-erlangen.de (W.L.); ulrich.rother@uk-erlangen.de (U.R.); 2Research Group GermanVasc, Department of Vascular Medicine, University Heart and Vascular Centre UKE Hamburg, University Medical Centre Hamburg-Eppendorf, 20246 Hamburg, Germany; ch.behrendt@uke.de

**Keywords:** tibial bypass surgery, CLTI, prosthetic bypass grafts, extra-anatomic bypass reconstruction

## Abstract

(1) Background: While tibial bypass surgery still plays a role in the treatment of patients with chronic limb-threatening ischemia and diabetic foot syndrome; only a few centers have recorded considerable numbers of these conditions. The current study aimed to determine contemporary practice with special focus on the performance of extra-anatomic grafting to the infrapopliteal arteries. (2) Methods: A retrospective, single-center study included patients with tibial bypass grafts from 1 January 2008 to 31 December 2019. Primary endpoints were complication rate, graft patency, amputation, overall survival, and major adverse cardiac (MACE) or limb event (MALE). The cohort was stratified by extra-anatomic vs. anatomic position. (3) Results: A total of 455 patients (31% female) with Rutherford stage 4 (12.5%) and 5/6 (69.5%) were included (thereof, 19.5% had high amputation risk according to the Wound Ischemia Foot Infection score). Autologous reconstruction was performed in 316 cases, and prosthetic reconstruction in 131 cases, with a total of 51 (11.2%) extra-anatomic grafts. Early occlusion rate was 9.0% with an in-hospital overall mortality of 2.8%. The in-hospital rate of MACE was 2.4% and of MALE, 1.5%. After one, three and five years, the primary patency of venous bypasses was 74.5%, 68.6% and 61.7%, respectively. For prosthetic grafts, this was 55.1%, 46.0%, and 38.3%, respectively (*p* < 0.001). The patency of extra-anatomic prosthetic grafts performed significantly better compared with anatomically positioned prosthetic grafts (log-rank *p* = 0.008). In multivariate analyses, diabetes (hazard ratio, HR 1.314, CI 1.023–1.688, *p* = 0.032), coronary artery disease (HR 1.343, CI 1.041–1.732, *p* = 0.023), and dialysis dependency (HR 2.678, CI 1.687–4.250, *p* < 0.001) were associated with lower odds of survival (4) Conclusion: In this large, single-center cohort, tibial bypass surgery demonstrated satisfactory results with overall low perioperative complication rates and long-term patency rates of 60% and 38%, respectively. Extra-anatomic bypasses represent a feasible alternative to venous grafts in terms of patency. A tailored, patient-centered approach considering predictors such as diabetes, dialysis dependency, and coronary artery disease along with prediction models may further improve the long-term results in the future.

## 1. Introduction

Bypass grafts to below the knee and tibial arteries have been used for the treatment of chronic limb-threatening ischemia (CLTI) since the 1970s [1,2]. However, even if superior long-term patency rates were reported for infrapopliteal saphenous vein grafts in comparison to all other types of interventions, comparative studies are scarce. To date, the quality of evidence is low, and the literature is outdated in parts [3]. Furthermore, suchlike analyses are oftentimes biased on distinct anatomic premises such as, for example, the treatment of significantly longer lesions in bypass surgery, compared to endovascular therapy (EVT) [4]. Due to the increasing numbers of diabetic patients and patients with end-stage renal disease (ESRD), treatment of the infrapopliteal vessels is required in a rising percentage of patients with CLTI in order to obtain direct blood flow to the pedal arteries [5]. In a recent prospective cohort study including 1676 patients with CLTI who underwent invasive revascularization in Germany, almost half of these patients had severe systemic disease or even a constant threat to life. The prevalence of ESRD was 5.4%, and 33% had chronic kidney failure [6]. Whereas the implantation of vein grafts represents the first-line therapy in this patient subgroup [7], the role of prosthetic grafts in this anatomic region is under debate due to inferior patency and limb salvage [8], which were reported to be as high as 40% in a 5-year follow-up [9]. Nevertheless, this approach still has its value in contemporary vascular surgery and is widely performed for CLTI in the absence of appropriate autologous graft material in the case of long-segment occlusions of the infrainguinal vessels or reocclusions after endovascular treatment [10]. Against this background, we report our 10 years’ experience in tibial bypass surgery; to the best of our knowledge, this series is the largest in terms of patient numbers published so far. The aim of the present study was to provide an overview on contemporary short- and long-term outcomes of tibial bypass grafts in order to facilitate decision making for surgical revascularization. Special focus was placed on the performance of extra-anatomic bypass reconstructions with prosthetic grafts and to detect prognostic factors for patency and limb salvage.

## 2. Materials and Methods

### 2.1. Patients

A retrospective, single-center analysis of consecutive patients treated with tibial bypass reconstructions due to CLTI or acute limb ischemia (ALI) at a university hospital was performed. Between 1 January 2008 and 31 December 2019, patients were screened for meeting the inclusion criteria. The study was conducted in congruence with the declaration of Helsinki, adhered to the STROBE (Strengthening the Reporting of Observational Studies in Epidemiology (STROBE) guidelines [11], and was further approved by the local ethics committee (number 40_21 Bc). Written informed consent was not necessary due to the study’s retrospective character.

### 2.2. Inclusion and Exclusion Criteria

Patients treated for CLTI or ALI by tibial bypass reconstructions were included in this study. During the index operation, the reconstruction was either performed by using the great saphenous vein or a prosthetic graft as bypass material. Additionally, the reconstruction could be performed either in an anatomic or an extra-anatomic manner. Excluded from this study were patients receiving prior endovascular revascularization methods as well as those who already had previous tibial bypass reconstructions of the index leg (tibial redo operations).

### 2.3. Study Design and Clinical Parameters

Different clinical parameters were assessed in addition to the baseline values such as age in years, dichotomized sex, and comorbidities (current smoker, end-stage renal disease, arterial hypertension, history of diabetes longer than two years, and hyperlipidemia). All data concerning the index bypass operation (tibial bypass operation) were retrospectively collected, including the time point of operation, the bypass graft material (the choice of the bypass material was in all cases a vein first approach if sufficient material was available), recipient vessel of the distal anastomosis, as well as the anticoagulation regime after the bypass operation. It was further assessed whether the reconstruction methods were used in an anatomic or extra-anatomic fashion. The indication for each operation was recorded (ALI or CLTI, including different Rutherford stages). The severity of the CLTI was estimated by the Wound Ischemia, Foot Infection Score (WIFI-Score).

The outcome parameters were assessed additionally. Therefore, in-hospital revision operations as well as long-term outcomes were recorded by assessing endpoints such as amputation, amputation-free survival, survival, bypass occlusion, major adverse cardiac events (MACEs), and major adverse limb events (MALEs) as well as wound healing rates.

Data regarding different treatment options after bypass occlusion including bypass–thrombectomy, redo–bypass, endovascular recanalization, or catheter-directed thrombolysis were collected.

### 2.4. Data Collection

The study was performed in a retrospective manner. Therefore, data were obtained by analyzing the patient files. Additionally, each patient was invited to a follow-up investigation; in cases where this was not possible, the patient was contacted by a telephone call follow-up. For those patients who were not contactable by telephone, the general practitioner was contacted by telephone. Additionally, in all patients to whom contact was not made personally, the register of deaths was used to obtain exact information on death rates. Thus, a complete follow–up for the endpoint survival could be realized.

### 2.5. Statistical Analysis

All statistical calculations were performed using SPSS 21 (SPSS Inc., Chicago, IL, USA). In the case of normal distribution, the mean and standard deviation (SD) were used; in cases of skewed distribution, median values together with minima and maxima were used. For qualitative factors, absolute and relative frequencies were given. The comparison of two independent groups was performed using the chi-squared test, Fisher’s exact test, the Mann-Whitney U test, or a two-sample *t*-test, as appropriate. To investigate the differences in patency and survival rates, Kaplan-Meier curves and the log-rank test were used to estimate the time to occlusion between venous and prosthetic bypass grafts. A Cox proportional hazard regression model was conducted that included arterial hypertonia, diabetes, coronary artery disease, hyperlipidemia, smoking, dialysis, and sex for the endpoint survival. For all statistical tests, *p* < 0.05 was considered to show a statistically significant difference, as adaption for multiple testing became unnecessary.

## 3. Results

### 3.1. Patients and Procedure Characteristics

A total of 455 patients were retrospectively included in this study (141 women, 314 men, 73.8 years, 24–95). The mean follow-up time was 45.1 months (0–153, SD 35.41). The risk profiles, clinical stages, and comorbidities are shown in Table 1. All patients received tibial bypass operations. The indication for operation was ALI in 52 cases and CLTI in 403 cases (for different stages, see Table 1). In 316 procedures, the used graft material was GSV (including eight composite vein grafts), in 131 cases, prosthetic graft material was used, and in 8 cases, a composite graft out of vein and prosthetic material was used. The bypass was positioned anatomically in 404 cases and extra-anatomically in 51 cases. The analysis of the distribution of the comorbidities between anatomic and extra-anatomic bypass reconstructions showed no statistical differences, with the exception of the usage of prosthetic bypass grafts (see Table 1). The inflow artery was the most common femoral artery in all patients. The target artery was the anterior tibial artery in 32.7%, the posterior tibial artery in 20.2%, the peroneal artery in 26.4%, and the tibio–peroneal trunk in 14.9% of cases. The remainder comprised patients with simultaneous jump grafts to a combination of tibial vessels.

### 3.2. Early Outcomes

In-hospital complication rates were analyzed and categorized as early complications. In total, 41 patients suffered from an early graft occlusion (18 vein grafts, 21 prosthetic grafts, and 2 composite vein-prosthetic grafts), leading to seven major amputations in total. Postoperative complications needing surgical revisions occurred in 28 cases (17 bleedings and 9 wound infections). A total of 11 patients suffered from myocardial infarction, and 13 died during the first 30 postoperative days.

### 3.3. Survival

The mean survival time was 45.1 (0–153, SD 35.46) months. Concerning endpoint survival, diabetes (HR 1.314, CI 1.023–1.688, *p* = 0.032), CAD (HR 1.343, confidence interval (CI) 1.041–1.732, *p* = 0.023), and dialysis (HR 2.678, CI 1.687–4.250, *p* < 0.001) were found as independent predictors for reduced survival.

### 3.4. Comparison of Prosthetic and Venous Bypass Material

The mean bypass patency was 29.03 (0–152, SD 32.3) months. The mean leg salvage time was 34.27 (0–152, SD 33.9) months. By comparing the bypass patency depending on the used graft material, significantly reduced patency rates were found for prosthetic bypass grafts (mean patency times: prosthetic grafts 20.7 (0–121, SD 26.1) months, venous grafts 32.62 (0–152, SD 34.1) months, log-rank: *p* < 0.001; see Figure 1). Similar results were found for the limb salvage rate by comparison of the different bypass materials (mean time of limb salvage: prosthetic grafts: 25.81 (0–144, SD 27.9) months, venous grafts: 37.99 (0–152, SD 35.8) months, log-rank *p* < 0.001; see Figure 2).

### 3.5. Analysis of Extra-Anatomic Reconstructions

For prosthetic bypass grafts, an additional analysis of the used reconstruction method (anatomic vs. extra-anatomic) was conducted. Interestingly, the patency rate was significantly better for the extra-anatomic reconstruction method (mean patency times: anatomic 14.54 (0–101, SD 21.6) months, extra-anatomic 33.21 (0–121, SD 29.9) months, log-rank *p* = 0.008). Similar results were found for the limb salvage (limb salvage time: anatomic 19.75 (0–101, SD 23.7) months, extra-anatomic 37.78 (0–144, SD 31.8) months, log-rank *p* = 0.009).

## 4. Discussion

The present study was conducted in order to give an overview of contemporary practice in tibial bypass surgery. The recent literature on this topic is scarce and mainly dates back to the end of the previous century [12]. We could demonstrate a considerably low in-hospital mortality of 2.8% and acceptable surgery-associated complication rates within the first 30 days (6%) in a cardiovascular-compromised patient subset. The long-term results also underline the durability of tibial bypass grafting. In congruence with the previous literature, present data confirm the superior patency rates of autologous graft material compared to prosthetic grafts [13,14]. One–year primary patency of 74.5%, three–year patency of 68.6%, and a five-year patency rate of 61.7% in vein bypasses compare favorably against one–year primary patency of 55.1%, three-year patency of 46.0%, and a five-year patency of 38.3% in prosthetic grafts. Described patency rates are in congruence with studies published decades ago [9,12], indicating that propagated modifications in surgical techniques such as, for example, vein collars or patches at the site of distal anastomosis in prosthetic grafts did not result in improved patency or influence limb salvage over time [15]. Vein cuffs have been hypothesized to decrease intimal hypoplasia [16] but failed to show any benefit in randomized trials [17,18]. However, a recent retrospective study found that use of vein patches was associated with lower reintervention rates and a trend towards an increased patency [16]. Against the background of unclear evidence, no distal vein collars were used in femorocrural bypasses in the present study; all grafts were directly anastomosed to the recipient artery. Regarding postoperative anticoagulation in prosthetic grafts, the regimen was adapted individually according to lower leg outflow and the quality of pedal arch as well as flow velocity in intraoperative control angiography. Dependent on comorbidities, either oral anticoagulation or dual antiplatelet therapy was indicated at the performing surgeon’s discretion, when pedal arch was incomplete or low flow was seen in the completion angiogram. In highly selected cases, autologous tibial reconstructions were also performed in Rutherford stage III patients with debilitating claudication after a frustrate conservative approach by means of walking exercise.

In multivariate analysis, end-stage renal disease was found as an independent predictor of mortality in tibial bypass grafting; this further emphasizes the poor survival rates in dialysis patients with CLTI based on their extensive comorbidities and reduced general state at presentation. However, in our view, this should not result in therapeutic nihilism, as previous studies demonstrated that femoro–distal bypasses can be performed with sound outcomes in terms of amputation-free survival in this challenging cohort [4,19].

Another research question in this study was the performance of prosthetic grafts dependent on their anatomic positioning. As there is evidence of superior long-term outcomes compared to expanded polytetrafluorethylene (PTFE) [20], we used ring-reinforced, heparin–bonded grafts as standard graft material for reconstruction in all patients. The usage of 6 mm ring-reinforced grafts is performed as standard procedure at our institution in prosthetic bypass below the knee in order to avoid compression during flexion/extension, especially for subcutaneously tunneled extra-anatomic grafts. As a rule, prosthetic grafts to the posterior or peroneal artery were placed below the sartorius muscle and tunneled between the femur condyles in an anatomic fashion, whereas bypasses to the anterior tibial artery were subcutaneously tunneled via an auxiliary incision on the lateral thigh as a lateral extra-anatomic graft. This method of reconstruction was published in the 1980s by Stockmann [21], and different kinds of PTFE grafts have been evaluated in this position, and patency rates of 30–40% after 3 years have been reported [22]. In general, extra-anatomic bypass grafting is often performed as a bail-out or rescue procedure, especially in aortoiliac occlusive disease [23]. Within the present analysis, we compared the results of prosthetic grafts and found significantly improved results in terms of patency and limb salvage in extra-anatomic grafts with a one and three-year patency of 76.3% and 66.1% versus 43.3% and 34.5% in anatomic bypasses. Hence, the performance of these grafts almost equals the infrapopliteal usage of the greater saphenous vein. To date, the reasons for this finding are unclear and must be further evaluated by following analyses; however, one possible explanation could be the foot outflow situation, as these bypasses were only indicated and implanted in the case of a patent dorsalis pedis artery; this might represent an advantage over bypasses to the peroneal artery with only collateral perfusion to the foot. As a matter of course, bias in terms of follow-up cannot be excluded due to the study’s retrospective nature. Even if a direct comparison of anatomic and extra-anatomic positioning is not feasible in the present non-randomized, retrospective series, extra-anatomic lateral anterior grafts showed surprisingly good long-term results and may pose a reasonable alternative to autologous vein grafts in frail patients due to their reduced operation time and the low invasiveness of the procedure.

The role of endovascular versus surgery first in the infrapopliteal segment is still under discussion [3]. Whereas better results as to patency and short-term amputation-free survival for bypass grafts [24,25] have been described, similar long-term outcomes as to amputation–free survival are reported for both approaches [3]; nevertheless, such comparisons are biased by high heterogeneity in terms of patient and lesion characteristics and especially by the fact that no long-segment occlusions are included in endovascular therapy studies [3,4]. Against this background, we may conclude that an individualized approach is essential in these patients and that tibial and pedal bypass grafts remain crucial therapy tools in CLTI and in the presence of long-segment occlusions of the infrapopliteal arteries. The limitations of the present study are characterized by the retrospective, non-randomized, single–center design with all its known drawbacks, as this work was designed as an observational register study.

## 5. Conclusions

Tibial bypass reconstructions can be performed with low perioperative complication rates in contemporary practice. Five–year patency rates are 60% for vein and 38% for prosthetic grafts. Extra-anatomic prosthetic bypasses showed a 5-year patency rate of 54% and do represent a feasible alternative to vein grafts in terms of patency. An individualized approach is mandatory, whereas decision making should include dialysis and CAD as independent predictors of postoperative survival.

## Figures and Tables

**Figure 1 jcm-11-01237-f001:**
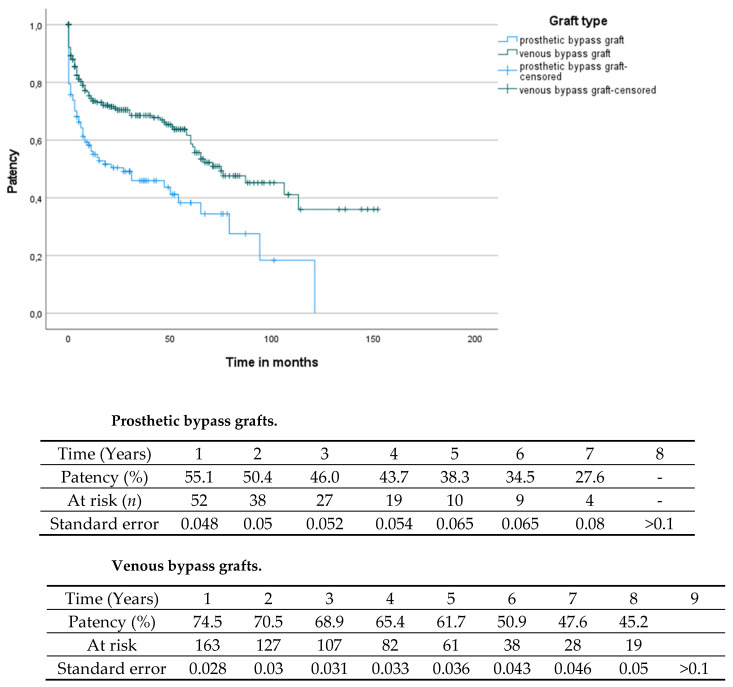
Comparison of the patency between prosthetic and venous bypass grafts (log-rank *p* < 0.001).

**Figure 2 jcm-11-01237-f002:**
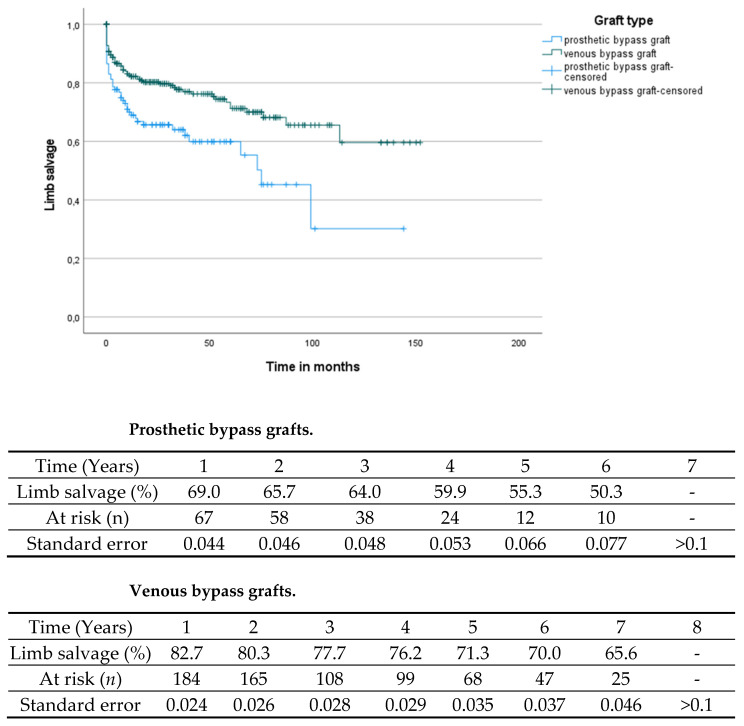
Comparison of the limb salvage rates depending on different bypass materials (log-rank *p* < 0.001).

**Table 1 jcm-11-01237-t001:** Patients and periprocedural characteristics and comparison of the distribution between the anatomic and extra-anatomic bypass groups.

Patients Characteristics	Total N (%)	Anatomic Bypass N (%)	Extra-Anatomic Bypass N (%)	*p*-Value
Arterial hypertension	412 (90.5)	365 (80.2)	47 (10.3)	0.804
Diabetes	229 (50.3)	208 (45.7)	21 (4.6)	0.183
Coronary artery disease	244 (53.6)	218 (47.9)	26 (5.7)	0.766
Hyperlipidemia	223 (49.0)	195 (42.9)	28 (6.2)	0.616
Smoking (currently and formal)	255 (56.0)	227 (49.9)	28 (6.2)	0.774
Dialysis dependency	25 (5.5)	22 (4.8)	3 (0.6)	0.904
**Procedure characteristics**				
Acute limb ischemia (as indication for bypass surgery)	53 (11.6)	45 (9.8)	8 (1.7)	0.353
Rutherford Stage				0.06
3	29 (6.3)	24 (5.3)	4 (0.09)	
4	57 (12.5)	47 (10.3)	10 (2.2)	
5/6	316 (69.5)	285 (62.6)	31 (6.8)	
Prosthetic bypass grafts	131 (28.8)	80 (17.6)	51 (11.2)	<0.001
Bypass anatomy	455 (100.0)	404 (88.7)	51 (11.2)	
WIFI Score amputation risk				0.451
Very low	9 (2.0)	9 (2.0)	0 (0.0)	
Low	76 (16.7)	63 (13.8)	13 (2.9)	
Moderate	90 (19.8)	81 (17.7)	9 (1.9)	
High	127 (27.9)	113 (24.8)	14 (3.1)	
Not assessable	153 (33.6)	135 (29.7)	26 (5.7)	
No antithrombotic therapy	12 (2.6)	9 (1.9)	3 (0.7)	0.091
Single antiplatelet therapy	218 (47.9)	194 (42.6)	24 (5.3)	
Dual antiplatelet therapy	46 (10.1)	41 (9.0)	5 (1.1)	
Vitamin K antagonist	119 (26.2)	106 (23.3)	13 (2.9)	
Direct oral anticoagulation	39 (8.6)	36 (7.9)	3 (0.7)	
Aspirin and direct oral anticoagulation	5 (1.1)	3 (0.7)	2 (0.4)	
Aspirin and Vitamin K antagonist	14 (3.1)	13 (2.9)	1 (0.2)	
Triple Therapy	2 (0.4)	2 (0.4)	0 (0.0)	

WIfI: Wound Ischemia foot Infection Score.

## Data Availability

Not applicable.

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
