# Peer review of "Long-Term Outcomes of Extra-Anatomic Femoro-Tibial Bypass Reconstructions in Chronic Limb-Threating Ischemia"

_jcm, 2022, doi:10.3390/jcm11051237_

Round 1

Reviewer 1 Report

This is an interesting article on femoro-tibial bypass reconstructions.

 Thank you for submitting this article. I was pleased to receive it as a reviewer.

The manuscript is generally well written, however I think that the statistical analysis has some shortcomings, I do not completely agree on certain sentences and I have some doubts about how the follow up was done.

A) Table.1

I don’t understand why the Authors put “Extra anatomic bypass” when it is the title of the column.

 B) Discussion

“Described patency rates are in congruence with studies published decades ago, indicating that propagated modifications in surgical techniques as for example vein collars or patches at the site of distal anastomosis in prostethic grafts did not result in improved patency or influence limb salvage over time”.

I don’t agree. I think they have to, at least, modify the sentence (see the article below).

Branco BC, Kougias P, Braun JD, Mills JL Sr, Barshes NR. Distal vein patch use and limb events after infragenicular prosthetic bypasses. J Vasc Surg. 2018 Jul;68(1):145-152. doi: 10.1016/j.jvs.2017.11.073. Epub 2018 Feb 10. PMID: 29439850.

C) Discussion

“..we use ring-reinforced heparin-bonded grafts as standard graft material for reconstruction in all patients”.

Why ring-reinforced? Please explain this choice

D)“..the performance of these grafts almost equals the infrapopliteal usage of the greater saphenous vein”

The Authors cannot write this sentence without a valid explanation. I think that it could be a bias in the follow up or in the statistical analysis.

Author Response

A) Table.1

I don’t understand why the Authors put “Extra anatomic bypass” when it is the title of the column

Response: We fully agree that is issue is misleading. WE substituted the caption "extra-anatomic bypass" for "bypass anatomy" in order to clarify and give the exact proportions of anatomic vs. extra-anatomic grafts. Compare Table 1

B) Discussion

“Described patency rates are in congruence with studies published decades ago, indicating that propagated modifications in surgical techniques as for example vein collars or patches at the site of distal anastomosis in prostethic grafts did not result in improved patency or influence limb salvage over time”.

I don’t agree. I think they have to, at least, modify the sentence (see the article below).

Branco BC, Kougias P, Braun JD, Mills JL Sr, Barshes NR. Distal vein patch use and limb events after infragenicular prosthetic bypasses. J Vasc Surg. 2018 Jul;68(1):145-152. doi: 10.1016/j.jvs.2017.11.073. Epub 2018 Feb 10. PMID: 29439850.

Response: thank you for this important comment. The corresponding statement was based on a prospective-randomized study performed on this topic several years before Branco et. al published their work; we modified the sentence according to your suggestion and included this citation in our references. Please compare references and page 6 line 195-197)

C) Discussion

“..we use ring-reinforced heparin-bonded grafts as standard graft material for reconstruction in all patients”.

Why ring-reinforced? Please explain this choice

Response: Usage of ring-reinforced grafts is performed as standard procedure at our institution in prosthetic bypass below-the- knee in order to avoid compression during flexion/extension especially for subcutaneously tunneled extra-anatomic grafts. This was added to the manuscript. Please compare page 7 lines 216-219

D)“..the performance of these grafts almost equals the infrapopliteal usage of the greater saphenous vein”

The Authors cannot write this sentence without a valid explanation. I think that it could be a bias in the follow up or in the statistical analysis

Response: This finding was in fact surprising. A possible explanation  could either be the outflow situation as corroborated in the manuscript or a bias in follow-up, as you correctly stated. We changed the wording accordingly. Please compare page 7 lines 236-237)

Reviewer 2 Report

I congratulate the group for an excellent review of the literature.  I have several questions and comments:

1) Was all inflow the common femoral artery?  A distal SFA to proximal PT has very different challenges/outcomes than a common femoral to distal PT.

2) where were the recipient arteries?  All at the origin of the vessel?  in other words, more granular data is required on where the bypass was implanted.  

3) Was a vein cuff used on the prosthetic bypass?  if so, what type?

4) what size prosthetic was used? 6mm?

5) I noticed that 29 bypasses were done for claudication - We rarely do a bypass for claudication to a single tibial vessel.  Can you please comment on why this was done?

6) What type of anticoagulation/anti-platelet therapy is used in prosthetic tibial bypasses? 

Author Response

1) Was all inflow the common femoral artery?  A distal SFA to proximal PT has very different challenges/outcomes than a common femoral to distal PT.

Response: The Inflow artery was the common femoral artery in all cases. This was added to the manuscript. (Compare page 3 line 135)

2) where were the recipient arteries?  All at the origin of the vessel?  in other words, more granular data is required on where the bypass was implanted.  

Response: Thank you for this remark. We included specific data on the corresponding target vessels, The individual site of anastomosis on the particular vessel was placed dependend on angiographic patency of the artery and the intraoperative level of calcification. Target artery was the anterior tibial artery in 32.7%, the posterior tibial artery in 20.2%, the peroneal artery in 26.4%, and the tibio-peroneal trunk in 14.9% of cases. The remainder comprised of patients with simultaneous jump grafts to a combination of tibial vessels. Compare page 3 lines 136-139)

3) Was a vein cuff used on the prosthetic bypass?  if so, what type?

Response: We clarified, that no vein cuffs were used in case of prosthetic grafts (Please see Discussion page 7 line 197)

4) what size prosthetic was used? 6mm?

Response; Yes indeed we exclusively used 6mm ring-reinforced grafts for prostetic revascularization of the below-knee arteries. This was completed in the manuscript. Compare page 7 lines 216-219)

5) I noticed that 29 bypasses were done for claudication - We rarely do a bypass for claudication to a single tibial vessel.  Can you please comment on why this was done?

Response: Tibial Bypass grafts in Rutherford III were only performed in case of debilitating claudication with long segment occlusion of popliteal and superficial femoral artery and these infrageniculate reconstructions were almost exclusively done with vein grafts; there must have been a strong therapeutic request of patients for walking distance, and walking exercise treatment for a 3 months period was mandatory before surgery, which was only indicated in case of frustrane conservative walking exercise. Compare page 7 lines 204-206)

6) What type of anticoagulation/anti-platelet therapy is used in prosthetic tibial bypasses? 

Response: Anticoaguation regimen is depicted in table 1. This was adapted individually according to lower leg outflow and the quality of pedal arch as wells as flow velocity in intraoperative control angiography. Dependent on comorbidities, either oral anticoagulation or dual antiplatelet therapy was indicated at performing surgeon's discretion, when pedal arch was incomplete or low flow was seen in completion angio. Compare page 7 line 199-204)

Round 2

Reviewer 1 Report

The Authors should report in the conclusion section the 5-year patency rate of the prosthetic extra-anatomic bypasses

Author Response

The Authors should report in the conclusion section the 5-year patency rate of the prosthetic extra-anatomic bypasses

Response: The 5-year patency rate of extraanatomic grafts was added to the conclusion section (Compare page 8 line 257)